# *Copy-Paste* Mutagenesis: A Method for Large-Scale Alteration of Viral Genomes

**DOI:** 10.3390/ijms20040913

**Published:** 2019-02-20

**Authors:** Jiajia Tang, Renke Brixel, Wolfram Brune

**Affiliations:** Heinrich Pette Institute, Leibniz Institute for Experimental Virology, 20251 Hamburg, Germany; jiajia.tang@leibniz-hpi.de (J.T.); renke.brixel@leibniz-hpi.de (R.B.)

**Keywords:** seamless mutagenesis, human cytomegalovirus, chimeric virus, vaccine

## Abstract

The cloning of the large DNA genomes of herpesviruses, poxviruses, and baculoviruses as bacterial artificial chromosomes (BAC) in *Escherichia coli* has opened a new era in viral genetics. Several methods of lambda Red-mediated genome engineering (recombineering) in *E. coli* have been described, which are now commonly used to generate recombinant viral genomes. These methods are very efficient at introducing deletions, small insertions, and point mutations. Here we present *Copy-Paste* mutagenesis, an efficient and versatile strategy for scarless large-scale alteration of viral genomes. It combines *gap repair* and *en passant* mutagenesis procedures and relies on positive selection in all crucial steps. We demonstrate that this method can be used to generate chimeric strains of human cytomegalovirus (HCMV), the largest human DNA virus. Large (~15 kbp) genome fragments of HCMV strain TB40/E were tagged with an excisable marker and cloned (copied) in a low-copy plasmid vector by *gap repair* recombination. The cloned fragment was then excised and inserted (pasted) into the HCMV AD169 genome with subsequent scarless removal of the marker by *en passant* mutagenesis. We have done four consecutive rounds of this procedure, thereby generating an AD169-TB40/E chimera containing 60 kbp of the donor strain TB40/E. This procedure is highly useful for identifying gene variants responsible for phenotypic differences between viral strains. It can also be used for repair of incomplete viral genomes, and for modification of any BAC-cloned sequence. The method should also be applicable for large-scale alterations of bacterial genomes.

## 1. Introduction

The *Herpesviridae* are a family of large double-stranded DNA viruses that replicate their genomes in the host cell nucleus. With genomes sizes ranging from 120 to 250 kbp the herpesviruses are among the largest viruses infecting vertebrates. The human herpesviruses comprise important and highly prevalent pathogens such as herpes simplex virus, varicella zoster virus, Epstein-Barr virus, human cytomegalovirus (HCMV), and Kaposi’s sarcoma-associated herpesvirus [1].

HCMV (human herpesvirus 5) is an opportunistic pathogen, which causes generally mild infections in healthy individuals, but is responsible for significant morbidity and mortality in immunocompromised individuals, particularly in hematopoietic stem cell and solid organ transplant recipients [2]. Moreover, HCMV transmission from mother to child during pregnancy is the most common congenital infection worldwide and causes long-term neurological damage in approximately 15% of congenitally infected infants [3].

HCMV has the largest genome of all human herpesviruses with a genome length of approximately 235 kbp and a coding capacity of at least 200 protein products and an even larger number of polypeptides [4,5,6,7]. The functions of most HCMV gene products and their roles in viral infection and pathogenesis are still unknown or incompletely understood, primarily because HCMV’s large genome size, slow replication kinetics, and cell association have been major obstacles to virus mutagenesis in cell culture. Traditionally, HCMV mutants were obtained by replacing a target gene with a selection marker by homologous recombination in permissive eukaryotic cells [8,9]. This procedure, which works reasonably well for some of the fast-replicating herpesviruses, proved to be time-consuming and inefficient when applied to HCMV. Hence only few recombinant HCMVs have been constructed with this method. The situation changed dramatically 20 years ago when the genomes of murine and human CMVs were cloned as bacterial artificial chromosomes (BACs) in *E. coli*. [10,11,12]. This has made the viral genomes amenable to the tools of bacterial genetics, which are much faster, more flexible, and easier to control than the traditional genetic recombination methods in eukaryotic cells [13,14,15]. Consequently, infectious BAC clones of many other herpesviruses, poxviruses, and baculoviruses have been generated and used for mutagenesis [16,17,18].

Although the first herpesvirus BAC mutants were constructed by homologous recombination using the *E. coli* RecABCD system [10,11,12] or by random transposon mutagenesis [19,20,21,22], the Red recombination system of bacteriophage λ quickly established itself as the most versatile and efficient system for recombination-mediated genetic engineering (recombineering). The λ Red recombination enzymes can be expressed in an inducible fashion from plasmid vectors [23,24] or from a defective prophage integrated in the *E. coli* genome [25]. The latter system, which allows a temperature-controlled expression of the λ Red recombinases, has become the most widely used system.

The λ Red recombination system initially required positive selection with an antibiotic resistance marker and was therefore most useful for the deletion of viral genes or the insertion of short sequences along with a selectable marker. However, the system was further developed to facilitate scarless removal of the selectable marker. This can be done either by combining positive and negative selection [26,27] or by flanking the positive selection marker with a short duplication on either side, which allows subsequent removal of the marker by recombination between the duplicated sequences [28,29]. The latter method of transient marker insertion has been termed *en passant* mutagenesis and has become one of the most widely used mutagenesis methods for BAC-cloned viral genomes.

Another application of the λ Red recombination system is for the subcloning of BAC fragments in plasmid vectors. This procedure has been called *gap repair* recombination and allows the cloning of BAC pieces up to 80 kbp in low-copy plasmid vectors [30].

While the methods described above are very efficient at introducing deletions, small insertions, and point mutations into BAC-cloned viral genomes, the insertion of larger sequences or the exchange of extended homologous sequences between viral strains (i.e., the construction of chimeric strains) has remained a challenge.

HCMV strains show a substantial genomic variability with a high number of single-nucleotide polymorphisms (SNPs) across the viral genome, many of which are coding relevant and thus affect the amino acid composition of viral proteins [4,31]. It is thus impossible to predict which of the many differences between strains are responsible for a particular strain-specific phenotype. The construction of chimeric viral strains by exchange of homologous sequences would therefore be instrumental for the identification of the genetic region (and ultimately the gene variant) responsible for a strain-specific phenotype. Moreover, chimeric strains could also serve as vaccine candidates [32]. Therefore we developed *copy-paste* mutagenesis, a cloning and recombination strategy that combines *gap repair* and *en passant* mutagenesis procedures. It allows scarless transfer of large fragments from one strain to another strain, thus generating chimeric strains. Using this method, we successively transferred four large fragments of HCMV strain TB40/E into strain AD169, thereby generating an AD169-TB40/E chimera containing 60 kbp of the donor strain TB40/E. The overall strategy of the *copy-paste* mutagenesis procedure is illustrated in Figure 1.

## 2. Results

### 2.1. Insertion of a Selectable Marker into the TB40/E Target Fragment

To generate chimeric HCMV strains consisting of AD169 and TB40/E sequences, we first divided both genomes into 15 large segments of approximately 15 kbp each. The TB40/E segments were named as A, B, C, D…O and their counterparts in AD169 were named as a, b, c, d…o (Figure 1). To replace the segment ‘a’ (UL112-127) in AD169 with the corresponding segment ‘A’ of TB40/E, we first inserted a kan^R^ selection marker into the fragment ‘A’ to allow positive selection during cloning and transfer to AD169 (Figure 2A). An I-SceI restriction site was inserted along with the kan^R^ marker to facilitate the scarless removal of the marker in the final step. The insertion was carried out in *E. coli* stain GS1783 using the *en passant* mutagenesis protocol [29]. GS1783 bacteria express the lambda Red recombination enzymes in a temperature-inducible fashion and the I-SceI restriction enzyme upon induction with arabinose [29]. The kan^R^/I-SceI cassette was PCR-amplified from plasmid pEPkan-S using oligonucleotide primers containing homology arms (HAs) to the target sequences in fragment ‘A’ and a short duplication for scarless excision later on. After purification, the PCR product was electroporated into recombination-competent GS1783 containing the TB40/E BAC (recombination-competent means that the bacteria had been prepared for electroporation and undergone a 12–15 min temperature shift for the induction of recombination enzymes). The kan^R^/I-SceI cassette was inserted into fragment ‘A’ by homologous recombination, and bacterial clones carrying the recombinant BAC were selected with kanamycin and chloramphenicol (resistance marker present on the BAC replicon). Successful insertion of the kan^R^/I-SceI cassette was expected to result in an altered restriction pattern of the viral genome. DNA from two clones (Ins3 and Ins4) were isolated and digested by three different restriction enzymes (Figure 2B). The digestions yielded restriction patterns similar to of those of the wildtype TB40/E BAC. The observed alterations were expected based on the known sequences of the TB40/E BAC and the kan^R^/I-SceI cassette. Thus we concluded that the kanamycin cassette was integrated correctly into region ‘A’. Clone Ins3 was used for the following experiments.

### 2.2. Cloning of Fragment ‘A-Ins’ in Plasmid pBR322

To isolate the fragment ‘A-Ins’ from TB40/E genome, we copied (i.e., cloned) it into the plasmid vector pBR322. This was done by using the *gap repair* method described previously [30] with a few modifications. The principle of the procedure is outlined in Figure 3A. Plasmid pBR322 was first linearized by digestion with restriction enzyme HindIII and then used as a template for PCR. Only the origin of replication (ori) and the ampicillin resistance (amp^R^) marker were amplified to minimize the size of the plasmid vector. HAs (50 bp each) homologous to fragment ‘A-Ins’ were introduced into the forward and reverse primers. Sequences of restriction enzymes SmiI and MssI were also introduced into the primers to allow excision of fragment ‘A-Ins’. After purification the PCR product was electroporated into competent GS1783 containing the TB40/E-A-Ins3 BAC. By *gap repair* recombination, a circular plasmid containing the pBR322 replicon and fragment ‘A-Ins’ was generated, and bacterial clones were selected with ampicillin and kanamycin. Plasmid DNA was then isolated from several clones and analyzed by restriction enzyme digestion (Figure 3B). Since the fragment ‘A-Ins’ has no restriction sites for SmiI and MssI, digestion of the plasmid with these enzymes yielded a 16 kbp fragment representing ‘A-Ins’ and a 2.7 kbp fragment representing the vector. Digestion of the plasmid DNA with HindIII or BamHI both resulted in the expected restriction patterns. Thus, the fragment ‘A-Ins3’ was successfully isolated from TB40/E and copied into plasmid pBR322.

### 2.3. Deletion of Fragment ‘a’ in AD169

Since genomes of different HCMV strains are highly homologous, fragment ‘a’ of AD169 had to be deleted before integration of ‘A’ to avoid unwanted recombination between ‘a’ and ‘A’. The deletion was done by replacing fragment ‘a’ with a zeocin resistance (zeo^R^) marker as shown in Figure 4A. The zeo^R^ marker was PCR-amplified with primers containing appropriate HAs, and the PCR product was electroporated into recombination-competent GS1783 bacteria containing the AD169 BAC. Bacterial clones containing recombinant BACs were isolated by selection with zeocin and chloramphenicol. The BAC DNAs of several clones were isolated and analyzed by restriction enzyme digestion. The results of two clones (Del3 and Del4) were shown in Figure 4B. The HindIII, EcoRI, and NdeI restriction patterns were consistent with expectations, indicating that fragment ‘a’ was successfully deleted in both clones. Thus, both clones were used for the following experiments.

### 2.4. Insertion of Fragment ‘A-Ins’ into AD169-a-Del

To insert fragment ‘A-Ins’ of TB40/E into AD169-a-Del, we first released ‘A-Ins’ from its plasmid vector by digestion with SmiI and MssI. The fragment was purified and electroporated into recombination-competent GS1783 containing the AD169-a-Del3 or the AD169-a-Del4 BAC. Successful homologous recombination should result in the integration of fragment ‘A-Ins’ and loss of the zeo^R^ marker (Figure 5A). Thus, bacterial clones were selected with kanamycin and chloramphenicol. To verify the loss of zeo^R^, bacterial clones were replica-plated on LB-agar plates with zeocin. Zeocin-sensitive clones were used for the second *en passant* mutagenesis step to remove the kan^R^/I-SceI cassette. As GS1783 express the I-SceI under the control of an arabinose-inducible promoter, addition of arabinose to the LB growth medium resulted in the expression of the I-SceI restriction enzyme and cleavage at the I-SceI recognition site within the kan^R^/I-SceI cassette. The free DNA ends generated by the cleavage stimulated homologous recombination between the short repeats flanking the kan^R^/I-SceI cassette, resulting in a scarless excision of the cassette. BAC DNAs of three clones (3-2, 4-1, and 4-2) were isolated and subjected to restriction enzyme analysis. Clones 3-2 and 4-1 both showed the expected pattern with digestion of HindIII and EcoRI (Figure 5B), indicating a correct integration of ‘A-Ins’ and removal of the kan^R^/I-SceI cassette. Clone 4-2 showed a slightly different pattern, suggesting that at least one of thehomologous recombination steps did not occur in the desired manner. In addition, both junctions were PCR-amplified and verified by sequencing. The two correct chimeric BAC clones were named AD169:A3-2 and AD169:A4-1.

Using the same strategy, three additional TB40/E fragments (‘B’, UL69-UL77; ‘C’, UL48A-UL55; and ‘D’, UL56-UL68) were introduced stepwise into AD169:A. The chimeric BACs were named AD169:AB, AD169:ABC, and AD169:ABCD, respectively. The HindIII and EcoRI restriction patterns of all four chimeras (two clones for each chimera, shown in Figure 6) matched the expected patterns. To further verify the correct assembly of the chimeras and exclude fortuitous mutations that might have occurred during the four rounds of *copy-paste* mutagenesis, the entire AD169:ABCD BAC was subjected to Illumina sequencing. The analysis of the sequence confirmed that the chimera had been constructed as intended and that there were no unwanted mutations.

### 2.5. Reconstitution of Chimeric Virus from BAC DNA

In order to test whether these chimeric constructs can generate infectious viruses, DNAs of these chimeric BACs were purified and transfected into MRC-5 human embryonic lung fibroblasts by electroporation. Infectious virus was recovered for all four chimeras. All chimeric viruses replicated well in human fibroblasts, and none of them showed an obvious growth defect. As an example we show the replication kinetics of chimera AD169:ABC compared to the parental wildtype AD169 strain (Figure 7).

## 3. Discussion

For a long time, the laboratory-adapted strains AD169 and Towne were the most widely used HCMV strains in experimental research. These strains are less cell-associated and grow to higher titers in cell culture as compared to clinical isolates. However, it has been known for many years that cell culture adaptation leads to the accumulation of point mutations, deletions, and occasionally even rearrangements [33,34]. These mutations can cause various phenotypic changes such as alterations in viral cell tropism [35,36] or resistance to apoptosis [37]. Besides the laboratory strains AD169 and Towne [11,12,38], several clinical strains have been cloned as infectious BACs [4,39,40,41]. As these strains were cloned after a limited number of passages in cell culture, they usually retained the broad cell and tissue tropism and other properties of the original virus. However, even short-term passage in cell culture can lead to mutations [41]. Moreover, increasing evidence indicates that clinical HCMV isolates can have divergent properties that may be relevant for pathogenesis in the human host. Thus, efficient methods for large-scale alteration of viral genomes will be required to identify genetic differences responsible for strain-specific properties.

The first method used for the construction of chimeric HCMVs was the cosmid recombination system. In this system a set of eight or more cosmids containing overlapping HCMV genome fragments are transfected into human fibroblasts where a full replication-competent viral genome is regenerated by homologous recombination [42]. This method has been used to generate a set of four chimeras of the HCMV strains Towne and Toledo, which were used as vaccine candidates [32]. Unfortunately, the cosmid recombination system is rather inefficient and difficult to handle and has therefore not been widely adopted. Moreover, complete sets of overlapping cosmid clones are not available for many of the commonly used HCMV strains.

In this study we present *copy-paste* mutagenesis as a new principle for the construction of chimeric HCMVs. The method combines *en passant* mutagenesis and *gap repair* cloning, two well-established and widely adopted methods. The method relies on homologous recombination and is therefore independent of the limitations of traditional molecular cloning such as availability of restriction sites and PCR size limits. All recombination steps were carried out by Red recombination in *E. coli* strain GS1783 and relied on positive selection, which ensured a high efficiency. Although only a few clones were analyzed for each step, the majority of them yielded the desired results.

In this study we transferred four 15-kbp fragments from TB40/E to AD169. We have not tried larger fragments, but are convinced that this should be possible. The crucial size limiting steps are *gap repair* cloning and purification of the cloned fragment (steps 2 and 3 in Figure 1). While *gap repair* has been shown to work for fragments up to 80 kbp [30], the purification of intact fragments might be more challenging when they are very large.

The use of *copy-paste* mutagenesis is not restricted to the construction of chimeric virus strains, but should also be useful for the repair of incomplete BACs or for the insertion of large fragments into unrelated BACs. In these cases, a homologous sequence is not present in the acceptor BAC making the deletion step (step 4 in Figure 1) unnecessary. Moreover, *copy-paste* mutagenesis should also be applicable for large-scale alterations of the *E. coli* genome.

Two other methods for large fragment recombineering have been published: the *ALFIRE* method [43] and the *landing pad* method [44]. These methods differ from *copy-paste* mutagenesis in that they rely on both positive and negative selection. Which of the three methods is better suited or more efficient for a specific application can only be decided once these methods have been used side-by-side. However, one advantage of the *copy-paste* method is that it avoids the difficulties associated with negative selection.

## 4. Materials and Methods

### 4.1. Plasmids and Reagents

The pCGN71 plasmid expressing the HCMV tegument protein pp71 has been described [45]. The cloning vector pBR322 was a gift from Marion Ziegler (Heinrich Pette Institut, Hamburg, Germany). The pEPkan-S plasmid [28] was provided by Nikolaus Osterrieder (Free University Berlin, Germany), and the pEM7/Zeo plasmid was from Invitrogen. *E. coli* strain GS1783 (DH10B λ*cI*857Δ(*cro-bioA*)<>*araC*-P_BAD_*I-sceI*) [29] was grown in LB broth (Lennox) containing 5 g/l NaCl (Sigma-Aldrich). For zeocin selection the pH was adjusted to 7.5. Antibiotics were purchased from Roth or Invitrogen and used at the following concentrations: ampicillin (100 µg/mL), kanamycin (50 µg/mL), chloramphenicol (15 µg/mL), and zeocin (25 µg/mL). L-(+)-arabinose was purchased from Sigma-Aldrich. Precisor DNA polymerase for PCR and buffer were purchased from BioCat, FastDigest restriction enzymes from ThermoFisher Scientific. Oligonucleotide primers were synthesized at Invitrogen. Primers used for transfer of fragment ‘A’ are listed in Table 1.

### 4.2. Cells and Viruses

MRC-5 human embryo lung fibroblasts (CCL-171) were obtained from the ATCC and grown in Dulbecco’s modified Eagle’s medium supplemented with 10% fetal calf serum, 100 U/mL penicillin, and 100 µg/mL streptomycin. The BAC clone of HCMV strain AD169*var*L [46] was a gift from Vu Thuy Khanh Le-Trilling (University of Duisburg-Essen, Germany). TB40-BAC4, the BAC clone of HCMV strain TB40/E, has been described previously [40]. To reconstitute infectious HCMV from BAC DNA, 3-5 µg BAC DNA together with 1.5 µg pCGN71 were electroporated into 10^7^ MRC-5 cells suspended in 500 µL Opti-MEM-I (Invitrogen, Carlsbad, CA, USA). Electroporation was carried out in a 4 mm cuvette using a BioRad GenePulser Xcell at 220 V and 950 µF. After 5 min, cells were flushed out of the cuvette with 1 mL OptiMEM-I. Coagulated debris was carefully removed and the remainder was transferred to a cell culture dish containing 10 mL growth medium. Cells were cultured until all cells showed cytopathic effects. Virus propagation and stock production was also done in MRC-5 cells. Viral titers were determined by using the median tissue culture infective dose (TCID_50_) method [47]. For viral growth kinetics, cells were infected in six-well dishes. The inoculum was removed after 2 h, cells were washed with PBS, and new medium was added. Supernatants were collected at different times post infection and stored at −80 °C for later titration.

### 4.3. BAC Mutagenesis

BAC mutagenesis was carried out in *E. coli* strain GS1783. Deletion of target sequences and replacement with a zeo^R^ marker was done essentially as described [48]. The zeo^R^ gene was PCR-amplified from pEM7/Zeo using primers containing 50-nucleotide HAs to the target region. Recombination-competent GS1783 containing the AD169 BAC were electroporated to introduce the PCR-generated fragment and initiate recombination. Bacterial clones carrying the recombinant BAC were obtained by plating on LB agar plates containing chloramphenicol and zeocin.

Insertion of kan^R^/I-SceI cassette was done according to the first step of the *en passant* mutagenesis protocol [29]. The cassette was PCR-amplified from pEPKan-S using primers containing 60-nucleotide HAs to the target region and a short (40 nucleotides) duplication for scarless excision later on. Recombination-competent GS1783 containing TB40-BAC4 were electroporated to introduce the PCR-generated fragment and initiate recombination. Bacterial clones carrying the recombinant BAC were obtained by plating on LB agar plates containing chloramphenicol and kanamycin.

Large fragments cloned by *gap repair* were excised from the pBR322 plasmid backbone by digestion with SmiI and MssI and purified by electrophoretic separation and gel extraction using a NucleoBond Gel and PCR Clean-up kit (Macherey-Nagel, Düren, Germany). Recombination-competent GS1783 containing the AD169 BAC were electroporated to introduce the purified fragment and initiate recombination. Bacterial clones carrying the recombinant BAC were obtained by plating on LB agar plates containing chloramphenicol and kanamycin. Bacterial colonies were replica-plated on zeocin plates to confirm the loss of the previously introduced zeo^R^ marker.

Finally, the kan^R^/I-SceI cassette was removed by the second step of the *en passant* mutagenesis procedure [29]. Selected clones were cultured in LB medium containing chloramphenicol at 30 °C for 2–3 h until the medium became cloudy. L-(+)-arabinose was added to the culture to induce the expression of I-SceI enzyme and cleavage of the BAC DNA at the I-SceI site. After 1 h, the culture was transferred to 42 °C to initiate recombination. After 13 min, the culture was transferred back to 30 °C for at least 1h. Bacterial clones carrying the recombinant BAC were obtained by plating on LB agar plates containing chloramphenicol and L-(+)-arabinose. Bacterial colonies were replica-plated on kanamycin plates to confirm the loss of the previously introduced kan^R^/I-SceI marker.

### 4.4. Gap Repair Cloning

Large fragments of TB40-BAC4 were cloned in plasmid pBR322 by using the *gap repair* method described previously [30] with a few modifications. The low-copy plasmid pBR322 was first linearized by digestion with HindIII. The replicative sequences and the amp^R^ marker were PCR-amplified using primers containing approx. 20 nucleotides of homology to pBR322 and 40 nucleotides of homology to the TB40-BAC4 fragment to be subcloned. Unique restriction enzyme sites (SmiI and MssI) were also introduced through the primers. The PCR product was then purified by gel electrophoresis and gel purification. Purified vector DNA (150 ng) was used to transform recombination-competent GS1783 containing TB40-BAC4 with a previously kan^R^/I-SceI-tagged region. Bacterial clones carrying the recombinant plasmid were obtained by plating on LB agar plates containing kanamycin and ampicillin. DNA of positive recombinants was isolated using mi-Plasmid Miniprep Kit (Metabion, Planegg, Germany) and examined by digestion with SmiI+MssI, HindIII, and BamHI.

### 4.5. BAC Sequence Analysis

The complete sequence of the AD169:ABCD BAC was determined with an Illumina MiSeq sequencer and analyzed as described [49]. The complete genome sequence of TB40-BAC4 is available at GenBank (accession no. EF999921). The complete sequence of the AD169*var*L BAC was kindly provided to us by Vu Thuy Khanh Le-Trilling (University of Duisburg-Essen). It is almost identical to the AD169*var*UC sequence (accession no. FJ527563).

## Figures and Tables

**Figure 1 ijms-20-00913-f001:**
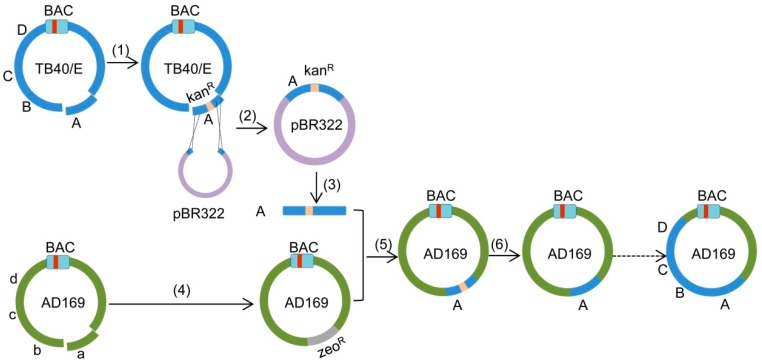
Schematic view of the *copy-paste* mutagenesis. (1) Insertion of a kan^R^/I-SceI cassette into fragment ‘A’ (UL112-127) of TB40/E by Red recombination, corresponding to the first step of *en passant* mutagenesis. (2) Copying of fragment ‘A-Ins’ into pBR322 by *gap repair* recombination. (3) Release of fragment ‘A-Ins’ from plasmid by restriction enzyme digestion. (4) Deletion of fragment ‘a’ (corresponding to fragment A in TB40/E) in AD169 by Red recombination. (5) Insertion of fragment ‘A’ into AD169-Del by Red recombination. (6) Removal of the kan^R^/I-SceI cassette by Red recombination, corresponding to the second step of *en passant* mutagenesis. The red bar within the bacterial artificial chromosome (BAC) cassette represents the chloramphenicol resistance (cam^R^) marker.

**Figure 2 ijms-20-00913-f002:**
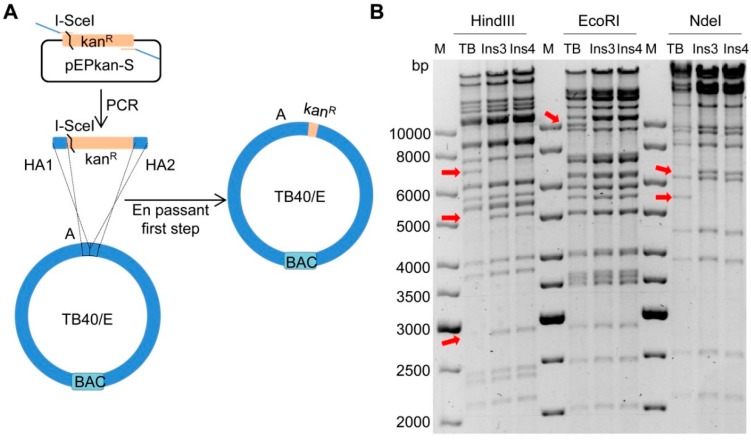
Insertion of a kan^R^/I-SceI cassette into fragment A of TB40/E. (**A**) Schematic view of the insertion strategy (first step of *en passant* mutagenesis). The kan^R^/I-SceI cassette was PCR-amplified using plasmid pEPKan-S as a template and primers containing homology arms (HA) homologous to the target sequences in fragment ‘A’. The purified PCR product was transformed into competent GS1783 carrying the TB40/E BAC. Recombinant BAC clones were selected by chloramphenicol and kanamycin. (**B**) BAC DNAs of two clones (Ins3 and Ins4) were analyzed by restriction digestion with HindIII, EcoRI, and NdeI, respectively. Arrows indicate the expected alterations caused by the insertion of the kan^R^/I-SceI cassette compared to the parental TB40/E genome (TB).

**Figure 3 ijms-20-00913-f003:**
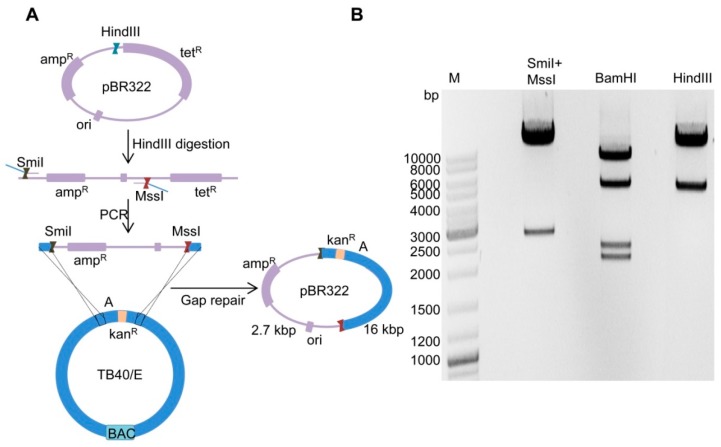
Cloning of the TB40/E ‘A-Ins’ fragment. (**A**) Schematic view of cloning by gap repair. Plasmid pBR322 was linearized by digestion with HindIII and used as template for PCR. Homology arms and restriction sites for SmiI and MssI were included in the primers. The PCR product was purified and electroporated into recombination-competent GS1783 carrying TB40/E-A-Ins3 BAC. Clones were selected with ampicillin and kanamycin. (**B**) Restriction digest of a positive plasmid DNA clone digested with SmiI+MssI, BamHI, and HindIII, respectively.

**Figure 4 ijms-20-00913-f004:**
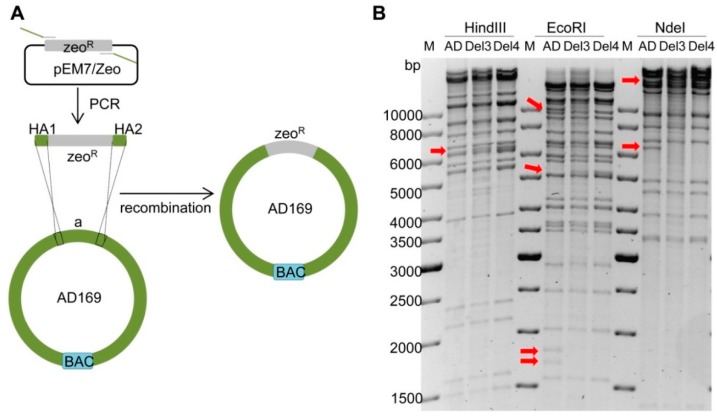
Deletion of fragment ‘a’ in AD169. (**A**) Schematic view of deletion strategy. The zeo^R^ gene was PCR-amplified with primers containing homology arms (HA) homologous to the targeted sequence of AD169. The PCR product was purified and electroporated into recombination-competent GS1783 containing the AD169 BAC. Recombinant clones were selected with zeocin and chloramphenicol. (**B**) BAC DNA was analyzed by restriction digestion with HindIII, EcoRI and NdeI. Arrows indicate the expected alterations in ‘a’ deleted genomes (Del3 and Del4) compared to the parental AD169 (AD) BAC. M, molecular weight standard.

**Figure 5 ijms-20-00913-f005:**
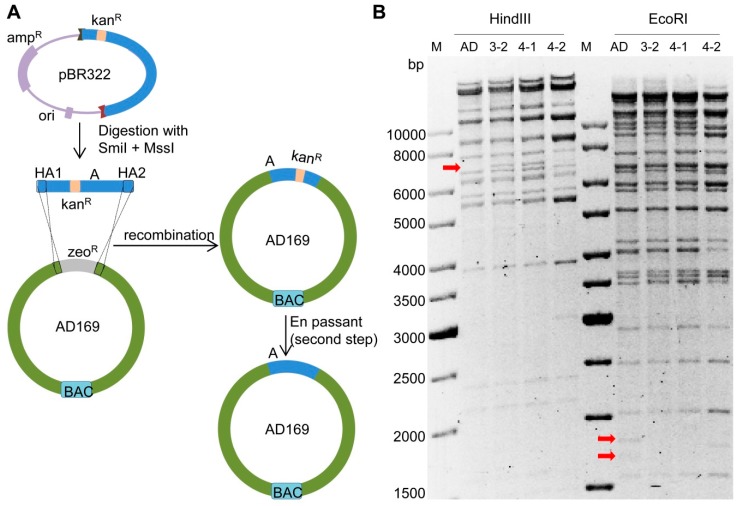
Insertion of fragment ‘A-Ins’ into AD169-a-Del. (**A**) Schematic view of the insertion (paste) strategy. Fragment ‘A-Ins’ was released by digestion with SmiI+MssI. Then it was purified and transformed into competent *E. coli* strain GS1783 carrying AD169-a-Del3 BAC or AD169-a-Del4 BAC. Clones were selected with kanamycin and chloramphenicol. The kan^R^/I-SceI cassette was removed by the second step of *en passant* mutagenesis. (**B**) BAC DNA was analyzed by restriction digestion with HindIII and EcoRI. Arrows indicate the expected alterations in the ‘A’ inserted genome (clones 3-2, 4-1 and 4-2) compared to the parental AD169 (AD) BAC. M, molecular weight standard.

**Figure 6 ijms-20-00913-f006:**
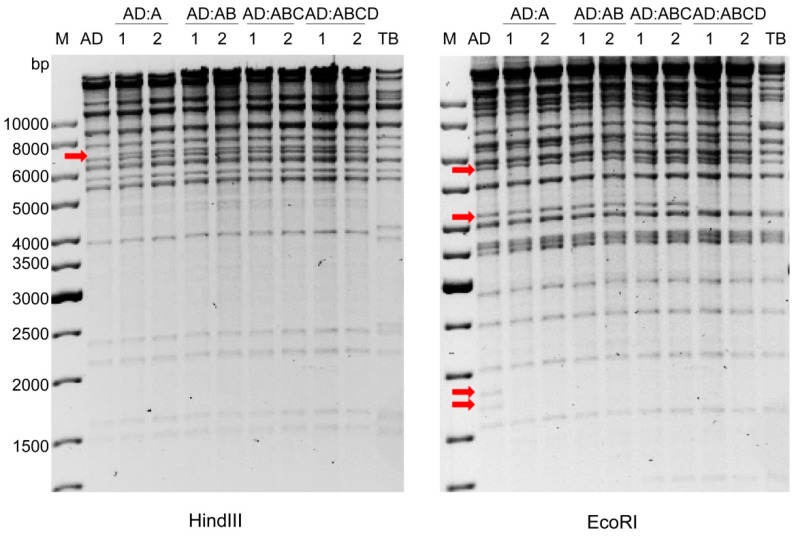
Restriction enzyme analysis of chimeric HCMV BACs. BAC DNAs of AD169:A, AD169:AB, AD169:ABC, and AD169:ABCD (two clones each) were isolated and analyzed by digestion with HindIII and EcoRI, respectively. Arrows indicate alterations in the restriction patterns of chimeric constructs compared to the parental AD169 (AD) genome. M, molecular weight standard; TB, TB40/E.

**Figure 7 ijms-20-00913-f007:**
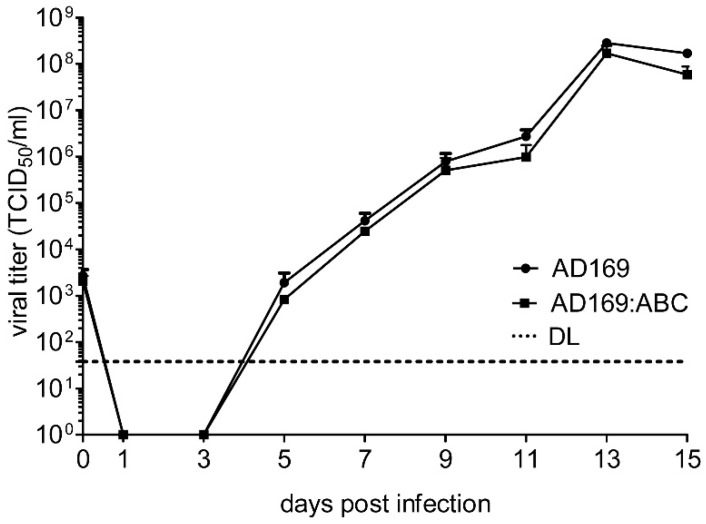
Multistep replication kinetics of chimeric AD169 in fibroblasts. Human embryonic lung fibroblasts (MRC-5) were infected with AD169:ABC or wildtype AD169 at an MOI of 0.03 TCID_50_/cell. Virus released into the supernatant was determined by titration. Means and SD of triplicates are shown. DL, detection limit.

**Table 1 ijms-20-00913-t001:** Primers used to transfer TB40/E fragment ‘A’ to AD169.

Primers	Sequences *	Purpose
A-MssI-Fwd	*CCCCGTAAACGATATAAGCGCTATCGCCAGATATCGCGTA* gtttaaac**GATACGCGAGCGAACGTGA**	Cloning of UL112-127 of TB40/E in pBR322
A-SmiI-Rev	*AAACTACGTCACCCGACACGCGGAAAAGAAAGACCGTCGC* atttaaat**TTCTTAGACGTCAGGTGGCAC**	
a-Del-Fwd	*TCCTCTTGTAGCAACGTGAGGACGACTACTCCGTGTGGCTCGACGGTACG* **TGTTGACAATTAATCATCGGCAT**	Deletion of UL112-127 in AD169 and replacement with zeo^R^
a-Del-Rev	*GTGTGTCGCAAATATCGCAGTTTCGATATAGGTGACAGACGATATGAGGC* **TCAGTCCTGCTCCTCGGCCA**	
A-Ins-Fwd	*CCATTTACCGTAAGTTATGTAACGCGGAACTCCATATATGGGCTATGAACTAATGACCCC* **TAGGGATAACAGGGTAATCGATTT**	Insertion of kan^R^/I-SceI cassette into UL112-127 of TB40/E
A-Ins-Rev	*GACATTGATTATTGACTAGTTATTAATAGTAATCAATTACGGGGTCATTAGTTCATAGCC* **GCCAGTGTTACAACCAATTAACC**	

* Homologous sequences are in italics, restriction sites in lower case, and PCR binding sequences in bold.

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
