# Peer review of "Copy-Paste Mutagenesis: A Method for Large-Scale Alteration of Viral Genomes"

_ijms, 2019, doi:10.3390/ijms20040913_

Round 1
Reviewer 1 Report
The manuscript by Jiajia Tang et al. describes a very elegant new method to exchange large DNA fragments between bacterial artificial chromosomes (BAC). The authors apply their new technology to generate chimeric human cytomegaloviruses (HCMV).
The manuscript is very well written.
The paper is highly relevant and the method will most likely be adapted by several groups working with BACs - in particular in the field of (herpes- and pox-) virology.
I just have some minor (nitpicking) points or suggestions:
- In the pdf version, which I received, the placement of the figures (preceding the corresponding text) was not immediately intuitive to me (see e.g. line 102 and Fig. 2).
- For readers beyond the field of CMV research, it might help to inscribe the chloramphenicol resistance gene in the BAC cassette in the figures.
- In the text, the authors use the word scarless. As keyword they use seamless. Both are fine, but maybe the text and the keywords can be ‘harmonized’.
- The authors use herpesviruses as plural. In my opinion, in line 41 it should be herpesviruses as well.
- In line 43, the authors refer to the complexity of the HCMV translatome/proteome. Maybe, the most recent paper by Erhard and Dölken dealing with this aspect might be also cited.
- In line 57, Dong Yu’s paper dealing with Tn mutagenesis of HCMV might be also cited (PMID 14519856).
- It might be worthwhile to briefly explain what ‘recombination-competent’ (see line 127 and the legend of Fig. 3) means in terms of GS1783 treatment and expression of Red and I-Sce.
- In line 149, the authors refer to the ori as replicon, which is correct but a bit unusual. In the figures, the term ori is used.
- In Fig. 7, descriptive statistics would be more appropriate in my opinion. Thus, SD should be used instead of SEM.
Author Response
We thank this reviewer for his positive feedback and constructive comments.
Point 1. In the pdf version, which I received, the placement of the figures (preceding the corresponding text) was not immediately intuitive to me (see e.g. line 102 and Fig. 2).
Response: In the revised version of the manuscript, the figures have been placed after the corresponding text.
Point 2. For readers beyond the field of CMV research, it might help to inscribe the chloramphenicol resistance gene in the BAC cassette in the figures.
Response: The chloramphenicol resistance marker within the BAC cassette is now shown in Figure 1 and described in the figure legend.
Point 3. In the text, the authors use the word scarless. As keyword they use seamless. Both are fine, but maybe the text and the keywords can be ‘harmonized’.
Response: We used 'seamless' on purpose as a key word that is not in the abstract. It helps colleagues to find the publication in PubMed, no matter whether they search for scarless or seamless mutagenesis. Both expressions have been used.
Point 4. The authors use herpesviruses as plural. In my opinion, in line 41 it should be herpesviruses as well.
Response: This has been corrected.
Point 5. In line 43, the authors refer to the complexity of the HCMV translatome/proteome. Maybe, the most recent paper by Erhard and Dölken dealing with this aspect might be also cited.
Response: Thank you for pointing that out. The reference to the Erhard and Dölken paper has been included (line 43).
Point 6. In line 57, Dong Yu’s paper dealing with Tn mutagenesis of HCMV might be also cited (PMID 14519856).
Response: We included this reference as well (line 57).
Point 7. It might be worthwhile to briefly explain what ‘recombination-competent’ (see line 127 and the legend of Fig. 3) means in terms of GS1783 treatment and expression of Red and I-Sce.
Response: A sentence explaining ‘recombination-competent’ has been included (lines 117-119).
Point 8. In line 149, the authors refer to the ori as replicon, which is correct but a bit unusual. In the figures, the term ori is used.
Response: ‘replicon’ was replaced by ‘origin of replication (ori)’ (line 144).
Point 9. In Fig. 7, descriptive statistics would be more appropriate in my opinion. Thus, SD should be used instead of SEM.
Response: Figure 7 was changed as requested. SD is now shown instead of SEM.
Reviewer 2 Report
In this study, Tang et.al. developed an an efficient and versatile method called Copy-Paste mutagenesisa for scarless large-scale alteration of viral genomes. They used this novel method to generate efficiently the chimeric strains of human cytomegalovirus (HCMV), the largest human DNA virus. This method is pretty useful for identifying gene variants responsible for phenotypic differences between viral strains and for repair of incomplete viral genomes, and for modification of any BAC-cloned sequence. Generally, the figures are easy to follow and data are well organized. However, there is only one mistake needing to be addressed before acceptance.
In the figure legend of Figure 5, the subtitle of (a) and (b) should be corrected into (A) and (B).
Author Response
Point 1. In the figure legend of Figure 5, the subtitle of (a) and (b) should be corrected into (A) and (B).
Response: The figure legend has been corrected as requested. Thank you for catching this!